# A Morphological and Size-Based Study of the Changes of Iron Sulfides in the Caples and Torlesse Terranes (Otago Schist, New Zealand) during Prograde Metamorphic Evolution

**Victor Cardenes** [1,2,*] **, Raúl Merinero** [3] **, Álvaro Rubio-Ordoñez** [1] **, Veerle Cnudde** [2] **, Javier García-Guinea** [4] **and Iain K. Pitcairn** [5]

1. Geology Department, Oviedo University, C/Jesús Arias de Velasco s/n, Oviedo, 33005 Asturias, Spain; rubioalvaro@uniovi.es
2. Pore-scale Processes in Geomaterials Research Team (PProGRess), Ghent University, Krijgslaan 281 (S8), 9000 Ghent, Belgium; Veerle.Cnudde@UGent.be
3. Mineralogy and Petrology Department, Complutense University of Madrid, Avda. Complutense s/n, 28040 Madrid, Spain; rmeriner@ucm.es
4. National Museum of Natural History, Jose Gutierrez Abascal 2, 28006 Madrid, Spain; guinea@mncn.es
5. Department of Geological Sciences, Stockholm University, SE-10691 Stockholm, Sweden; iain.pitcairn@geo.su.se
* Correspondence: cardenesvictor@uniovi.es

**Abstract:** It is widely accepted that metamorphism induces a remobilization of iron sulfides, sweeping away original ones while creating new ones. This paper analyzes size distributions of iron sulfides in several samples from the Caples and Torlesse terranes from the Otago Schist (New Zealand) using high-resolution X-ray computed tomography, which allows all iron sulfides larger than the resolution at which X-ray scans were performed to be characterized. Framboids and clusters of framboids are common in unmetamorphosed samples, but disappear in greenschist/amphibolite facies samples, where iron sulfides have anhedral habits. By contrast, the size and standard deviation of the new iron sulfides both remain within the same range. The results illuminate the evolution of iron sulfides throughout metamorphism, proposing boundaries for the metamorphic processes based on the shape of these iron sulfides.

**Keywords:** micro-pyrite; framboids; micro-tomography; size distribution; morphological evolution

---

## 1. Introduction

Microscopic iron sulfides (mainly pyrite and pyrrhotite, together with marcasite and greigite) can be found in many geological environments under a wide range of morphologies, from framboids to anhedral and euhedral crystals. Almost ubiquitous in sedimentary and metamorphic environments, iron sulfides can provide an accurate record of the many geological processes that affected the rock during its geological history. Moreover, iron sulfides are known to be efficient scavengers of trace elements that, in many cases, have a significant economic value. Many papers have studied the genesis, geochemical evolution, geological meaning and occurrence of iron sulfides. However, the turning points in this line of research are two studies by Wilkin and co-workers [1,2] who proposed a feasible model for framboidal pyrite formation. They also proposed a method to infer paleo-redox conditions using the size distributions of framboids. In the years since, other authors have continued working with populations of framboidal pyrite [1–7] applied to the determination of paleo-conditions. Of the diverse

factors controlling framboid growth and morphology, the most important are reactant concentration, temperature and time [1–9].

There are three main approaches to the study of iron sulfides: isotope analysis, trace element analysis, and the previously mentioned size distribution characterization. Isotope analysis generally focuses on the relationships between sulfur isotopes, as well as other isotopes found in the host rock. Trace element analysis is useful for charting the remobilization of elements throughout diagenesis and metamorphism. Finally, size distribution techniques are based on the study of the size and abundance of the micro-pyrites (MPy). Isotope and trace element analysis provides information about geochemistry, while size distribution is useful in order to infer the paleo-conditions. This latter method, size distribution analysis, mainly uses SEM to characterize the MPy populations. However, there are two issues related to this technique: the representativeness of the volume scanned and the underestimation of the true diameter, as it involves measuring a 3D object via a 2D surface. Recently, high-resolution X-ray computed tomography (MCT) [10], together with statistical analysis based on Gaussian finite mixture models, have been used to characterize the size and abundance of iron sulfide populations [11–14]. This method allows the populations of iron sulfides to be measured in 3D, in a fast and accurate way, preventing possible errors due to the 2D observation of sections as performed in SEM studies. Although for some authors this difference amounts to less than 10% [15], the truth is that comparing SEM versus MCT for MPy determination [12] yielded 1513 objects determined by SEM and 14,170 objects determined by MCT, which in fact amounts the difference up to 90%.

Iron sulfides are suggested to undergo a change in size, shape and distribution as one passes from diagenesis to metamorphism, together with changes in mineralogy and isotopic relationships [16–20]. This size increase can be the result of the growth of new iron sulfides, or of the aggregation of the existing iron sulfides. In a recent paper [13] we have shown that low-grade metamorphism prompts the development of new populations of framboids in pelitic rocks as metamorphic conditions change. The transition from diagenetic to metamorphic conditions creates new populations of framboids [12], which have the same morphology as sedimentary framboids [21] and therefore cannot be distinguished from them. A proper statistical analysis would highlight these new populations and mark the transition between diagenesis and metamorphism. However, these studies involve a certain degree of uncertainty regarding the samples' composition, as the rocks used are from different formations, which could influence the results.

The present study uses selected samples from the previous research of Pitcairn et al. 2010 [22], in the Caples and Torlesse terranes from the Otago and Alpine Schists in New Zealand, which are a metasedimentary pelitic sequence exhibiting unmetamorphosed greywackes up to rocks equilibrated under amphibolite facies [23]. These terranes have different provenance. Caples terrane is located to the south and west of the Otago Schist, and it is composed mainly of volcanoclastic greywacke, while Torlesse terrane, located to the north and east, is formed by quarzofeldespathic greywacke and argillite [24]. This sequence constitutes an exceptional test bench for metamorphic petrologists, since it makes it possible to observe the effects of prograde metamorphism on a single lithology. The Otago Schist is part of a Mesozoic metasedimentary belt that formed during the collisions of the Torlesse and Caples terranes in the Mesozoic. The schists comprise dominantly meta-greywackes and metapelites. The metamorphic grade increases from unmetamorphosed rocks of the Torlesse and Caples terranes on the flanks of the belt, to greenschist facies rocks in the central part. In [22], samples were analyzed for their geochemical compositions using low detection limit techniques and results show that a specific suite of elements including Au, As, and Sb are progressively mobilized from the rock during prograde metamorphism from unmetamorphosed graywacke to amphibolite facies condition. Driving forces for element mobility are temperature and progressive change from trace element-rich diagenetic pyrite that dominates in low metamorphic grade conditions to trace element poor pyrrhotite that is the dominant sulfide in upper greenschist and amphibolite facies. These sulfide minerals undergo systematic changes in size and texture during metamorphism that are similar to those recorded in other areas [17,20,25,26]. However, to our knowledge, these changes have not been quantified previously. Our work takes as its

starting point the textural observations and selected samples from [22] to evaluate the evolution of the size and texture of iron sulfides over the course of prograde metamorphism using high-resolution X-ray computed tomography with statistical analysis of log-normal populations, complemented by SEM observations.

## 2. Materials and Methods

High-resolution X-ray computed tomography makes it possible to distinguish rock-forming minerals depending on their attenuation coefficient to X-radiation. Each mineral has a different attenuation coefficient (Figure 1), which can be represented in a greyscale image of the sample. Representative samples for each metamorphic stage (Table 1) from the Torlesse and Caples terranes were selected.

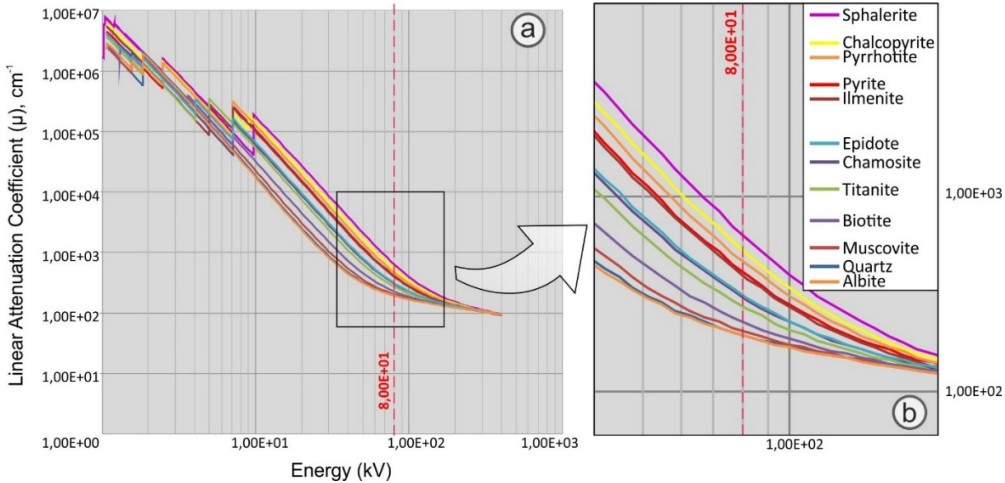

**Figure 1.** Linear attenuation coefficients of the minerals found in the samples. The voltage of the X-ray tube was 130 kV.

**Table 1.** Samples selected from [22]. GW: Greywacke; Psa: Psammite; Pel: Pelite; QFS: Feldspar quartzite; un-met: unmetamorphosed; GS: Greenschist grade. The population analysis shows the following data for the sub-populations: mean, standard deviation (SD), percentage of objects (%n) and percentage of volume (V) from the total population. Bottom rows show total volume and number of objects for each sample.

| Sample | A6 | A4 | A5 | C46 | A1 | A3 | C56 | C72 |
|---|---|---|---|---|---|---|---|---|
| Terrane | Caples | Caples | Caples | Torlesse | Torlesse | Torlesse | Torlesse | Torlesse |
| Lithology | GW un-mt | Pel sb-GS | Pel sb-GS | GW un-mt | Psa sb-GS | Pel sb-GS | QFS GS | QFS GS |
| **Population Analysis** | | | | | | | | |
| P1(mean,SD) | 9.6(1.7) | 9.7(1.8) | 9.8(1.8) | 11.5(3.2) | 10.2(2.1) | 12.5(3.6) | 9.9(1.9) | 7.7(0.5) |
| (%n,V) | 31.6,2.5 | 22.3,0.5 | 20.2,0.5 | 43.9,96.7 | 24.1,0.6 | 36.3,5.1 | 33.4,1.8 | 12.5,0.7 |
| P2(mean,SD) | 15.3(4.6) | 13.7(2.8) | 13.9(2.2) | 25.4(14.3) | 18.7(6.3) | 23.3(6.6) | 15.2(4.6) | 9.8(0.6) |
| (%n,V) | 45.1,15.4 | 22.7,1.4 | 20.0,1.2 | 56.1,97.7 | 34.4,5.7 | 52.1,40.1 | 34.4,7.0 | 20.7,1.8 |
| P3(mean,SD) | 27.7(14.7) | 23.8(6.2) | 19.5(2.9) | – | 40.3(21.5) | 44.7(12.7) | 30.2(16.6) | 11.7(1.1) |
| (%n,V) | 23.4,82.1 | 25.4,6.4 | 17.2,2.7 | – | 41.5,93.7 | 11.7,54.7 | 32.2,91.3 | 14.5,1.9 |
| P4(mean,SD) | – | 33.5(24.8) | 27.7(6.4) | – | – | – | – | 13.3(1.7) |
| (%n,V) | – | 29.6,91.6 | 18.2,8.3 | – | – | – | – | 14.1,2.8 |
| P5(mean,SD) | – | – | 25.7(14.4) | – | – | – | – | 20.0(6.8) |
| (%n,V) | – | – | 24.4,87.3 | – | – | – | – | 25.9,22.3 |
| P6(mean,SD) | – | – | – | – | – | – | – | 32.0(18.1) |
| (%n,V) | – | – | – | – | – | – | – | 12.2,70.5 |
| Tot.(mean,SD) | 27.7(14.7) | 21.9(0.6) | 24.3(18.9) | 19.5(14.5) | | 21.9(12.0) | 18.3(13.5) | 15.7(10.1) |
| (%n,V) | (100,100) | (100,100) | (100,100) | (100,100) | (100,100) | (100,100) | (100,100) | (100,100) |
| Vol. (mm³) | 43.1 | 152.8 | 204.0 | 114.2 | 190.2 | 41.1 | 45.8 | 194.1 |
| Objects | 4358 | 4219 | 5036 | 5623 | 4871 | 2568 | 2744 | 12611 |

For each sample, cylinders of 4 mm diameter × 8–10 mm length were taken. These cylinders were scanned at the HECTOR facility, a versatile research tomograph instrument located at the Centre for X-ray Tomography (UGCT) at Ghent University (Ghent, Belgium). Tube power was set to 10 W, with a high voltage of 130 kV. For each cylinder, over eighteen hundred projections were taken during a complete rotation along the vertical axis. The resolution achieved was 3 μm. Projections were rendered into a 3D volume using the program Octopus Reconstruction v.8.8.2. The resulting images were analyzed with the software ImageJ, obtaining datasheets corresponding to the size distributions. Using the methodology detailed in [14], these datasheets were then analyzed using the statistical software R v.3.4.3. and the mclust package v.5.4.1. The total size distribution is then separated into sub-populations that follow thereby locating the mixtures of sub-populations in the original size distributions. The applied methodology can discriminate between these sub-populations depending on geometrical parameters. The sphericity parameter was determined in order to differentiate between sub-populations based on their morphology, using the program ImageJ v.1.52i with the plugins 3D Fast filters and RoiManager3D v.3.92. The 3D Fast filters obtain descriptive statistical data in 3D (3D average, median, minimum, maximum, etc.), while RoiManager3D can determine the geometrical parameters (volume, sphericity, ellipsoid fitting, etc.) used to segregate the MPy subpopulations. More details about this process can be found in [11–14]. According to [27], an average sample volume scanned by MCT is much larger than the representative elementary volume (REV). This REV depends on the occurrence and size of the measured property. For MPy, previous works [12–14] have concluded that the REV is achieved when the number of objects is above 600–1000. The size and distribution of the MPy can be adjusted to a log-normal distribution, according to the Crystal Size Distribution (CSD) and the law of proportionate effect [28], which explains how a crystal population tends to grow in proportion to its size. Several authors have demonstrated that MPy populations follow log-normal distributions [1,29].

In order to complement the size population analyses, SEM observations were carried out using a Hitachi TM3000 Tabletop Microstructure, with an EDS Bruker Quantax 70 analyzer. The observations were carried out on thin polished sections of the samples. This equipment is located at Gea Asesoría Geológica (www.geaasesoriageologica.com) in Asturias, Spain.

## 3. Results and Discussion

As pointed out above, MCT distinguishes objects based on the difference of their attenuation coefficient, which depends in turn on their mineralogical composition. The greater this difference, the higher the contrast will be. Our samples exhibit a great deal of mineralogical diversity, making them more difficult to study. The main minerals [22] are silicates (quartz, plagioclase, K-feldspar and phyllosilicates), with some accessory minerals like ilmenite, zircon and titanite. The sulfides are pyrite (up to 0.1%) and pyrrhotite (up to 0.4%), with small percentages of chalcopyrite, sphalerite, galena and cobaltite. The abundance of these minerals depends on the specific composition and metamorphic grade of each sample. While the overall mineralogy allows for effective differentiation between the main minerals and iron sulfides, the occurrence of accessory minerals with similar sizes to the iron sulfides could introduce a miscalculation in the MCT results. However, the filtering method used is capable of segregating objects by shape, removing minerals with morphologies that do not match the target objects. The only mineral that could induce this error is ilmenite, if shape analysis does not allow its differentiation from iron sulfides, because it has a rather similar attenuation coefficient to iron sulfides (Figure 1). However, ilmenite is only found in unmetamorphosed greywackes, being replaced by titanite during further metamorphism, and therefore its abundance was considered not to be significant enough to alter the results. The morphological and statistical analyses showed no notable evidence in this respect.

Following the methodology of [14], a general population of spherical and isomorphic objects was obtained for each sample. Objects from the lower size intervals (2–16 μm) were filtered in order to reduce noise and artifacts of the equipment. The Caples and Torlesse terranes did not show significant differences in terms of lithology. Rather, differences are linked to the metamorphism.

For the unmetamorphosed samples, this general population clearly corresponded to a log-normal population for low and medium sizes (see QQ plots in Figure 2). In the sub-greenschist facies samples, this correspondence is less accurate, and greenschist-amphibolite facies samples lack such correspondence altogether. The analysis of the log-normal population mixtures (general populations) conducted using the mclust package indicated that the number of log-normal populations increases with increasing metamorphism (Figure 2). This is to be expected, according to the mineralogical composition of each sample as described in [22]. Pyrite is the most abundant sulfide in unmetamorphosed samples (C46 and A6). In sub-greenschist samples (A1, A3, A4 and A5) the abundance of pyrrhotite, and to a lesser extent chalcopyrite and sphalerite, increases. In samples from greenschist (C72) and amphibolite facies (C56), there is essentially no pyrite, sphalerite and galena. Only pyrrhotite and chalcopyrite are present, and therefore a higher number of log-normal populations of idiomorphic objects is to be expected in the mixture of the general population. These mineralogical changes are therefore the result of metamorphic recrystallization where new mineralogical phases form from the previous ones.

The correspondence between the log-normal populations detected with the mclust package and the spherical or idiomorphic objects can be achieved using SEM observations of each sample, and 3D reconstruction based on MCT images [14]. Figure 3 shows the main shapes detected with SEM and several 3D reconstructions of individual objects. In unmetamorphosed samples, only framboids and clusters of framboids were observed. Moreover, 3D reconstructions show individual framboids in smaller spherical objects.

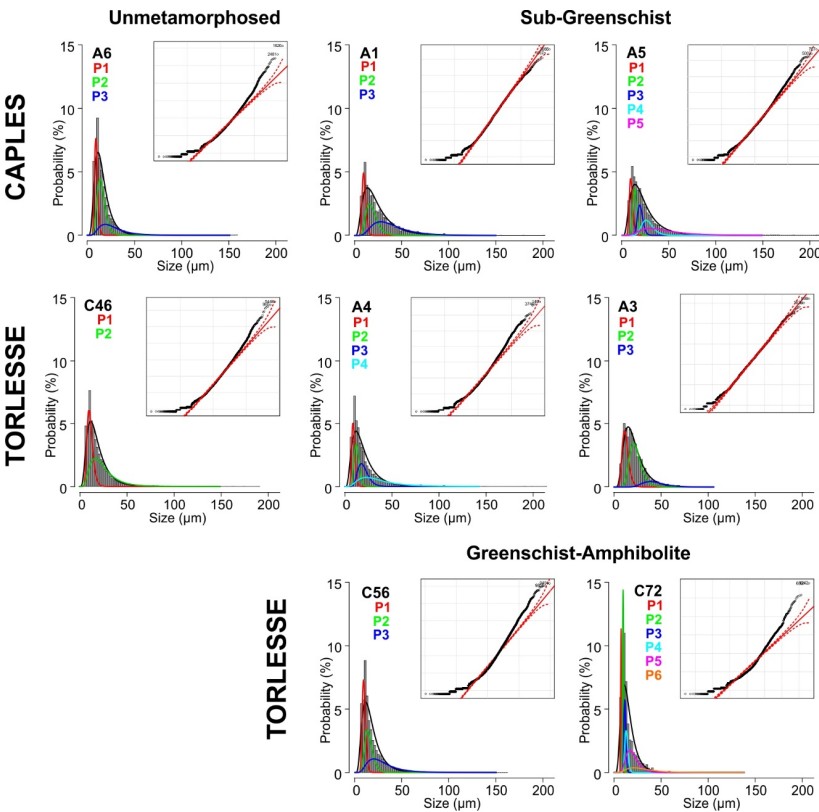

**Figure 2.** Population size distributions and QQ plots. For the population graphics, the black curve is obtained by high-resolution X-ray computed tomography (MCT), while color curves are the estimated with the statistical approach. QQ plots reflect the goodness of the fit of the calculated distribution (black curve) to the theoretical distribution (red curve).

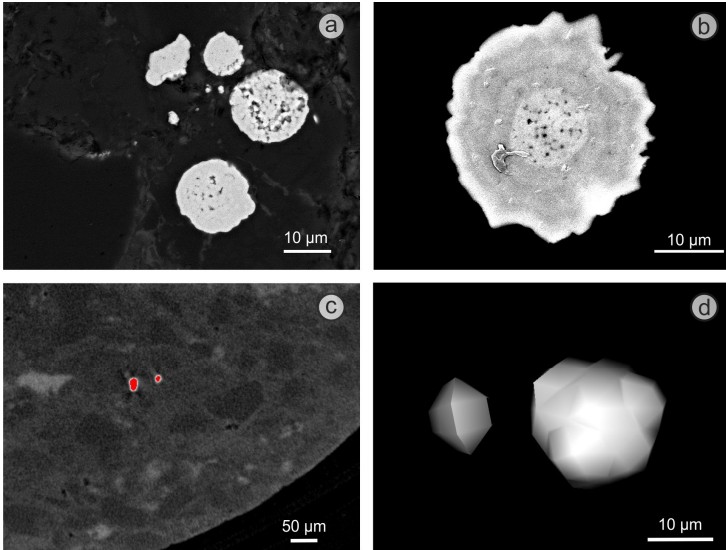

**Figure 3.** SEM and MCT images of selected samples. (**a**) Pyrite framboids, sample A6. (**b**) Pyrite framboid partially dissolved and then surrounded by recrystallization of pyrite. (**c**) Two iron sulfides differentiated from the rest of the sample during the threshold stage of the image analysis. (**d**) Reconstructions of two objects: left, a euhedral-like object, and right, a framboid-like object.

Therefore, log-normal populations, which make up the general population in unmetamorphosed samples, were interpreted as framboids and clusters. In sample A6, the second log-normal population interpreted as framboids, as well as the first population, probably grew out of previous framboids as metamorphism prograde. Similar interpretations can be made in samples from the sub-greenschist facies, where two populations of framboids corresponding to the lowest sizes are detected and log-normal populations of clusters are also present. The progradation of metamorphism promotes changes in the framboids' morphologies and can even bring about their outright destruction. In samples from the greenschist facies, the sulfides exhibit regular and rotated synkinematic textures in which the outer zones of the grains truncate the foliation, which explains why log-normal populations with other spherical shapes (euhedral) were detected.

The graphic representation of average size against the standard deviation of the populations of framboids constitutes a practical visual tool commonly used in paleoenvironmental studies in order to characterize the environment [3]. This representation is used here to compare the log-normal populations of iron sulfides detected in the analyzed metamorphic samples (Figure 4). In unmetamorphosed samples and samples from the sub-greenschist facies, framboids and clusters are dominant. Samples from the sub-greenschist facies show a slightly higher mean size for clusters, probably because during metamorphism, dissolution and recrystallization phenomena might involve the creation of new pore system with higher sizes. For samples from the greenschist and amphibolite facies, there is not a describable morphology and no framboids were detected. Based on the mean and standard deviation of the sizes plotting in the range of the detected framboids for unmetamorphosed and sub-greenschist facies samples, we may conclude that these unrecognizable morphologies were the consequence of the dissolution and/or transformation of earlier framboids. Framboids are always formed under oxic conditions during metamorphism, and probably other spherical shapes as euhedral crystals of pyrite were formed under anoxic conditions previous to the metamorphism, but no evidence of this can be established with the study of morphologies of sulphides with MCT. Finally, there is not a clear distinction between samples from the two terranes regarding subpopulations.

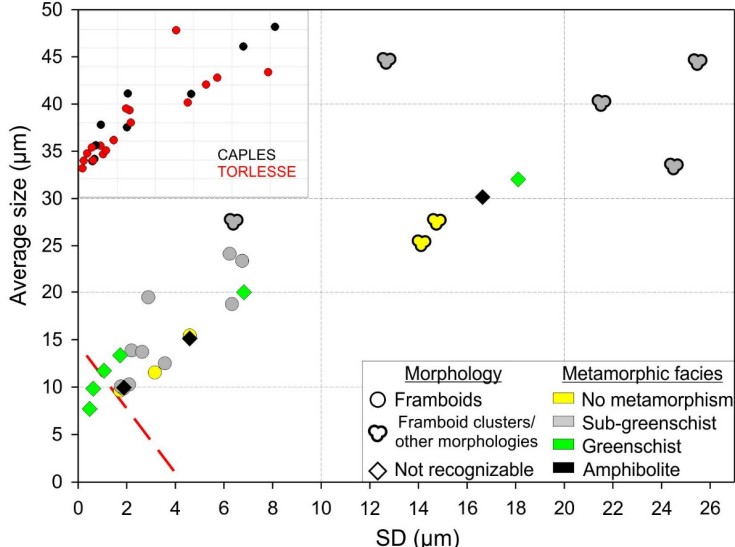

**Figure 4.** Graph of average size vs. standard deviation. The red dashed line separates the anoxic (bottom) and oxic (top) fields. This line is plotted as a reference, in order to compare this graph with others used in sedimentology and paleoenvironmental studies. The small graphic in the upper left corner shows samples from the Caples and Torlesse terranes.

## 4. Conclusions

Framboids and framboid clusters can be found in unmetamorphosed and sub-greenschist facies samples, but as metamorphism increases (greenschist and amphibolite facies) there is a transformation of the iron sulfides that leads to the erasure of these framboidal textures. Previous work has shown that the prograde transition from diagenetic to metamorphic conditions creates new populations of framboids [13]. This fact is confirmed here, where two log-normal populations of framboids were detected in samples belonging to the sub-greenschist facies. A step further in low-degree metamorphic conditions would create new iron sulfides formed without a defined morphology, but with an average size and standard deviation within the same range as framboids and clusters. Therefore, this transformation of framboids would specifically mark the transition between very low and low metamorphic conditions. It is important to remark that these results have been found in a pelitic sequence. For other type of rocks, these processes might be different. To clarify this point, further research is needed.

**Author Contributions:** Conceptualization, V.C. (Victor Cardenes), R.M. and I.K.P.; methodology, V.C. (Victor Cardenes), R.M. and V.C. (Veerle Cnudde); software, R.M. and Á.R.-O.; validation, R.M., J.G.-G. and A.R.; formal analysis, V.C. (Victor Cardenes) and Á.R.-O.; investigation, V.C. (Victor Cardenes); resources, A.R. and J.G.-G.; data curation, I.K.P.; writing—original draft preparation, V.C. (Victor Cardenes); writing—review and editing, V.C. (Victor Cardenes), R.M.; visualization, V.C. (Victor Cardenes); supervision, V.C. (Victor Cardenes); funding acquisition, Á.R.-O. All authors have read and agreed to the published version of the manuscript.

**Funding:** This work has been funded by the project Marie Curie IEF 623082 *TOMOSLATE*, granted by the European Research Council, and partially funded by the project PA-18-ACB17-11, from the Program Marie-Curie COFUND funded by the European Union, Government of Asturias (Spain) and the Spanish FICYT.

**Conflicts of Interest:** The authors declare no conflict of interest. The funders had no role in the design of the study; in the collection, analyses, or interpretation of data; in the writing of the manuscript, or in the decision to publish the results.

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
