# Peer review of "A Morphological and Size-Based Study of the Changes of Iron Sulfides in the Caples and Torlesse Terranes (Otago Schist, New Zealand) during Prograde Metamorphic Evolution"

_minerals, doi:10.3390/min10050459_

Round 1

Reviewer 1 Report

Dear Editor,

The manuscript by Cárdenes et alii is a short communication reporting the application of X-ray micro-tomography to the study of microscopic Fe-sulfides during prograde metamorphism. I think the text is well written and well organized and deserves publication in Minerals provided the authors consider the following main issues. 

Best regards

  1. The authors claim that size distribution studies of micro-sulfides based on SEM observations suffer from the problem of “representativeness of volume scanned”. However, in the ms, the real sample volume scanned by MCT is never clearly stated (“….cylinders of 4 mm were taken.” In addition the authors should add some comments about the fundamental issue of the representativeness of their sample volumes.  
  2. It is unclear to me, and likely to future readers inexpert of this kind of statistical processing, how the different sub-populations are extracted from the general population. Please clarify, if possible.
  3. The diagrams reported in Fig. 2 are too small and therefore not very useful. In addition they lack axis titles.
  4. It is important that authors be more cautious in generalizing their results. I am not a metamorphic petrologist, but I have observed both well-preserved framboidal pyrite in and euhedral pyrite micro crystals in meta-sedimentary rocks equilibrated under green schist facies.
  5. Caption of Figure 1: kV is not a unit of energy ! Please correct.

Additional corrections

- row 20: "... the evolution of the rocks’ size and morphology....". This phrase is unclear.

- row 34: "(mainly pyrite and pyrrhotite)" - why not marcasite ?

- row 38: please delete "rare and".

- row 54: ..."while size distribution is useful in order to infer the paleoconditions." This is also true for isotope and trace element data.

- row 79: please change "...up to amphibolite facies.." into ""...up to rocks equilibrated under amphibolite facies.".

- row 81: please delete "aged".

- row 89: "The driving force for....." . This sentence makes no sense. The driving force of the element mobility is the increase in temperature (mainly) ! 

- row 115: please change "psamite" with "psammite". In table 1 the abbreviation "Pel" is not defined.

- row 117: it is not clear if 4 mm is the radius, diameter or height of the cylinders.

- row 148: please add a comma beteween "ilmenite" and "if" and between "sulfides" and “because".

- row 151: please replace "recristallization to" with "replaced by"

- row 155: please quantify the "lower size intervals".

- row 167: please delete "..left in these higher grade samples”.

- row 184: please switch "left" and “right".

- row 203: "....probably due to the formation of higher pore sizes during metamorphism." Unclear, please explain.

- Figure 2: please add axis titles with units  to every diagram and units to the tables

- Figure 4: the red dashed line is of no use and confusing here. I would delete it. Please add unit (µm) to SD. In the legend replace ”Greenschists” and “Sub-greenschists” with ”Greenschist” and “Sub-greenschist”.

- References: I found several errors in the style of the entries. Please check

Author Response

  1. The authors claim that size distribution studies of micro-sulfides based on SEM observations suffer from the problem of “representativeness of volume scanned”. However, in the ms, the real sample volume scanned by MCT is never clearly stated (“….cylinders of 4 mm were taken.” In addition the authors should add some comments about the fundamental issue of the representativeness of their sample volumes.

The problem of “representativeness of volume scanned” is not related to the volume of the analyzed samples. It is related to the number of objects measured with the MCT technique in comparison with previous methodologies used to measure the diameter of framboids. In table 1 we include volumes, number of objects (MPy) analyzed and other data regarding the experimental conditions. The representativeness of the samples is also discussed in lines 133-140

  1. It is unclear to me, and likely to future readers inexpert of this kind of statistical processing, how the different sub-populations are extracted from the general population. Please clarify, if possible.

The statistical processing is based in previous works (reference [12-15]) and basically consists in separate into log-normal population the spherical objects measured with the MCT technique. A detailed explanation of this methodology here is not necessary. However, to give the reader a glimpse of the method, we include a short explanation (see lines  122-133)

  1. The diagrams reported in Fig. 2 are too small and therefore not very useful. In addition they lack axis titles.

We make the diagrams bigger, separating the tables and merging the results with Table 1, and finally adding axis titles. Please see modified Figure 2 and Table 1. In figure 2, size distributions have been enlarged, but QQ plots remain the same, since these graphics are used to make comparison between theoretical log-normal quantiles (axis X) and measured quantiles (axis Y).. The adjustment of measured data to theoretical log-normal is represented in the graphic with red (theoretical) and black (measured) lines. Figure caption is also updated.

However, the final size of the figure will be determined during the publishing process (in case the paper is finally accepted).

  1. It is important that authors be more cautious in generalizing their results. I am not a metamorphic petrologist, but I have observed both well-preserved framboidal pyrite in and euhedral pyrite micro crystals in meta-sedimentary rocks equilibrated under green schist facies.

Response: Framboidal and euhedral pyrite can be formed under many different environmental conditions, from euxinic sediments till greenschist facies. It is then common to find these microscopic pyrites in the rocks mentioned by the reviewer. In the introduction we state that these pyrites are practically ubiquitous in sedimentary and metamorphic facies.

To give an answer to the reviewer requirement, we add in the conclusions that this research has been done in a pelitic sequence, so our results are, in principle, valid for this type of rocks. See lines 239-241

  1. Caption of Figure 1: kV is not a unit of energy ! Please correct.

Response: We change kV by keV (sorry)

Additional corrections

- row 20: "... the evolution of the rocks’ size and morphology....". This phrase is unclear.

Indeed, the phrase is unclear and makes no sense. We believe it is a mistake from the translation process. Then we prefer to just delete it.

- row 34: "(mainly pyrite and pyrrhotite)" - why not marcasite ?

In our research, we have always found mainly these two forms, pyrite and pyrrhotite. However, other phases are found, such as greigite, marcasite and magnetite. We include these phases in the text, see lines 28-29

- row 38: please delete "rare and".

Done

- row 54: ..."while size distribution is useful in order to infer the paleoconditions." This is also true for isotope and trace element data.

True, we include this remark in the text

- row 79: please change "...up to amphibolite facies.." into ""...up to rocks equilibrated under amphibolite facies.".

Done

- row 81: please delete "aged".

Done

- row 89: "The driving force for....." . This sentence makes no sense. The driving force of the element mobility is the increase in temperature (mainly) ! 

We modify the sentence, including temperature as driving force, see lines 85-86

- row 115: please change "psamite" with "psammite". In table 1 the abbreviation "Pel" is not defined.

Done

- row 117: it is not clear if 4 mm is the radius, diameter or height of the cylinders.

We add “diameter”

- row 148: please add a comma beteween "ilmenite" and "if" and between "sulfides" and “because".

Done

- row 151: please replace "recristallization to" with "replaced by"

Done

- row 155: please quantify the "lower size intervals".

Done

- row 167: please delete "..left in these higher grade samples”.

Done

- row 184: please switch "left" and “right".

Done

- row 203: "....probably due to the formation of higher pore sizes during metamorphism." Unclear, please explain.

We change the sentence to “…during metamorphism, dissolution and recrystallization phenomena might involve the creation of new pore system with higher sizes” see lines 213-214

- Figure 2: please add axis titles with units  to every diagram and units to the tables

Done

- Figure 4: the red dashed line is of no use and confusing here. I would delete it. Please add unit (µm) to SD. In the legend replace ”Greenschists” and “Sub-greenschists” with ”Greenschist” and “Sub-greenschist”.

This red line is usually add in this type of graphic, which was firstly developed by sedimentologist and paleoenvironmentalist (e.g. Wilkin et al. 1996). We prefer to keep it so the graphic can be put in the same context than those from previous work, even when the topic is different. However, if after this explanation the reviewer considers that should be deleted, no problem.

We add the units for SD and replace the terms in the legend

Wilkin RT, Barnes HL, Brantley SL (1996) The size distribution of framboidal pyrite in modern sediments: An indicator of redox conditions. Geochim Cosmochim Acta 60 (20):3897-3912. doi:10.1016/0016-7037(96)00209-8

- References: I found several errors in the style of the entries. Please check

References were inserted and formatted using Endnote with the style provided by the journal. We have checked references, but we have not been able to find the errors in the style.

Reviewer 2 Report

I have read the paper “A morphological and size-based study of the changes of iron sulfides from the Otago formation (New Zealand) during prograde metamorphic evolution”

The material in this paper is of interest but the links between the XRT and SEM sample characterisations are rather poorly made.  I think this where this paper could provide some insight but it appears that framboids and framboid clusters are the only morphologies recognised.  The correlation between size and SD appears to be the main result of this work yet is not developed to any great extent.

General Comments

  1. Geologists are rather fastidious regarding the naming of rock units.  In the title “Otago formation” is used.  This is not a recognised name.  In the Abstract “Otago Schist Formation” is used.  This is also not a recognised name.  The term “Otago Schists” is used somewhat colloquially, but is inappropriate here because unmetamorphosed Torlesse and Caples rocks are analysed.  Suggest Mortimer and Roser (1992) https://doi.org/10.1144/gsjgs.149.6.0967 for terminology.
  2. I think the key element of this paper is somewhat missing.  It should be the connection between the X-ray tomography and the mineral morphology.  The link is sort of made in Fig. 3 but it doesn’t work for me.  I see these framboids and can see that they are framboids.  But, growth by augmentation and in particular epigenetic recrystallisation is not shown.
  3. The Caples terrane has units with a lot of volcanic pyrite.  No mention of this.  I suspect Torlesse and Caples will be distinct, but this is not addressed here at all (see Mortimer and Roser 1992).

Specific Comments

L16 First sentence of abstract is uninformative.  Suggest starting at second sentence.

L20 … evolution of the rocks’ size … I don’t think this is what you mean (pyrite size?)

L22 Otago Schist Formation.  There is no Formation with this name.

L34 Pyrite and Pyrrhotite not distinguished in this work

L53 Information about geochemistry vs paleo conditions - I don’t see the distinction.  Usually size distributions are used in terms of detrital minerals and understanding energy of deposition etc.  I’m not sure I’ve seen it used for metamorphic recrystallisation before.

L56 “yields sound results” - what does this mean? Judgement?

L59 Is 10% significant here?  I can see the benefits of XRT though in terms of speed and accuracy.

L70 new framboids from old.  This should be established for the samples you are looking at and they should be documented.  Granted it may only be morphology but this is useful information.

L75 The sources are indeed from two distinct rock types.  This should be better documented in results section.

L81 There are two distinct lithologies here.  Further, greywacke is a rather generic term.

L117 There are a lot of software packages mentioned here.  Would suggest putting their names in italics or something to allow their recognition.

L118 setup?  Facility?

L120 this description …

L121 I’m not an expert in XRT so why is the average voltage distinct from peak?

L131 I think the morphologies are key to making this a good paper.

L138 Samples show diversity.  This must be explained and the samples properly documented otherwise there is no point to this paper.

L151 recrystallisation: ilmenite - titanite is not a simple recrystallisation.  it is a metamorphic transformation involving movement of Si, Fe and Ca (for a start).

L170  Again, these “recrystallisations” are the result of chemical reactions

L179 Figure 2 is rather uninformative without descriptions of the parameters

L181 Figure 3 does indeed show framboids in (a) and (b).  Interesting that (b) appears to be structured suggesting Oswald ripening.  But what happens at higher grade?  Is there a morphological shift?  What is the point of (c)? Simply the recognition of pyrite in the algorithm? (d) interesting that the algorithm seeks to put everything in to euhedral crystals.  The framboid certainly doesn’t look like this.  And we see no comparator for the euhedral pyrite.

L189 metamorphism … ran its course.  Not sure what this means.

L192 … increase in metamorphic degree … needs to be more specific.

L194 “regular synkinematic textures”  euhedral? rotated?

L196 unrecognisable shapes (probably euhedral)… So euhedral shapes are not recognisable?  Look at the samples.

L205 there is not a recognisable morphology …  No.  Morphology can be described.

L208 We need more documentation of these unrecognisable features.

L211 Figure 4. I don’t understand the symbol distinctions here.  Looks like everything is either a framboid or a framboid cluster.  No formation of epigenetic pyrite?  There is no interpretation of the data on this plot.  Is the correlation between size and SD meaningful in any way? Is there a difference between Torlesse and Caples?

L217 if the framboids disappear, where does the Fe S go?  Note in Fig 3, sulphides are apparent in amphibolite grade.

L222 There is little discussion concerning the relationship between the size/SD and the mineralogy/morphology

Author Response

Open Review

English language and style

( ) Extensive editing of English language and style required
(x) Moderate English changes required
( ) English language and style are fine/minor spell check required
( ) I don't feel qualified to judge about the English language and style

Yes

Can be improved

Must be improved

Not applicable

Does the introduction provide sufficient background and include all relevant references?

( )

(x)

( )

( )

Is the research design appropriate?

( )

(x)

( )

( )

Are the methods adequately described?

( )

(x)

( )

( )

Are the results clearly presented?

( )

( )

(x)

( )

Are the conclusions supported by the results?

( )

( )

(x)

( )

Comments and Suggestions for Authors

I have read the paper “A morphological and size-based study of the changes of iron sulfides from the Otago formation (New Zealand) during prograde metamorphic evolution”

The material in this paper is of interest but the links between the XRT and SEM sample characterisations are rather poorly made.  I think this where this paper could provide some insight but it appears that framboids and framboid clusters are the only morphologies recognised.  The correlation between size and SD appears to be the main result of this work yet is not developed to any great extent.

General Comments

  1. Geologists are rather fastidious regarding the naming of rock units.  In the title “Otago formation” is used.  This is not a recognised name.  In the Abstract “Otago Schist Formation” is used.  This is also not a recognised name.  The term “Otago Schists” is used somewhat colloquially, but is inappropriate here because unmetamorphosed Torlesse and Caples rocks are analysed.  Suggest Mortimer and Roser (1992) https://doi.org/10.1144/gsjgs.149.6.0967 for terminology.

According to the reviewer´s suggestion, we use the nomenclature from the provided reference. Title is then changed to “A morphological and size-based study of the changes of iron sulfides in the Caples and Torlesse terranes (Otago schist, New Zealand) during prograde metamorphic evolution “. We also include a brief description of both terranes and the reference provided by the reviewer.

  1. I think the key element of this paper is somewhat missing.  It should be the connection between the X-ray tomography and the mineral morphology.  The link is sort of made in Fig. 3 but it doesn’t work for me.  I see these framboids and can see that they are framboids.  Growth by augmentation and in particular epigenetic recrystallisation is not shown.

The statistical analysis segregates MPy depending on the size and shape, using the software for image analysis ImageJ, together with the plugins 3D Fast Filters and ROI Manager 3D. This methodology allowed us to differentiate MPy morphologies present in the samples. On the other hand, using statistical analysis we can extract the subpopulations of MPy, i.e. we can segregate between “pure” MPy and recrystallized MPy. So MPy subpopulations generated by different genetic mechanisms (augmentation and epigenetic recrystallization) are statistically inferred. This methodology is explained in Merinero et al. 2017 (reference number 15).

However, it is important for the reader to understand the methodology, so we have modified the text. Please see lines 126-140

  1. The Caples terrane has units with a lot of volcanic pyrite.  No mention of this.  I suspect Torlesse and Caples will be distinct, but this is not addressed here at all (see Mortimer and Roser 1992).

The Caples and Torlesse have different provenence - more volcanogenic material in the Caples.  In our samples there is no evidence of detrital pyrite - all the sulfide in these samples is diagenetic.  Both the Caples and Torlesse have layers that are sulphide rich as is common in sedimentary sequences that include shaley layers.  Both the Torlesse and Caples contain metavolcanic layers but these are not common (5% of the total metasedimentary package) and none of the samples in this study are metavolcanic.

Specific Comments

L16 First sentence of abstract is uninformative.  Suggest starting at second sentence.

Done

L20 … evolution of the rocks’ size … I don’t think this is what you mean (pyrite size?)

Indeed, the phrase is unclear and makes no sense. We believe it is a mistake from the translation process. Then we prefer to just delete it.

L22 Otago Schist Formation.  There is no Formation with this name.

Changed

L34 Pyrite and Pyrrhotite not distinguished in this work

We just mention these minerals in the introduction. However, these samples are the same than those from Pitcairn et al. 2010, were this distinction is done

L53 Information about geochemistry vs paleo conditions - I don’t see the distinction.  Usually size distributions are used in terms of detrital minerals and understanding energy of deposition etc.  I’m not sure I’ve seen it used for metamorphic recrystallisation before.

Size distributions and its evolution through metamorphism have been mentioned in several papers (e.g. Vokes 1969; Tempelman-Kluit 1970; Craig 1983; McClay & Ellis 1983; Sassano & Schrijver 1989; Craig & Vokes 1993; Craig et al. 1998; Large et al. 2007; Scott et al. 2009). Since 2016, we have been applying high definition X-ray tomography to analyze these size distributions (Cardenes et al. 2019, Merinero et al. 2019, Merinero et al. 2017, Cardenes et al. 2016), something that had never been done before due to the limitations of the techniques used so far (SEM). This methodology is a novelty, so perhaps this is the reason the reviewer has not seen it before.

L56 “yields sound results” - what does this mean? Judgement?

We mean that results are solid, but certainly the sentence is not clear, it is better to delete “yields sound results”, since it does not affect the general meaning.

L59 Is 10% significant here?  I can see the benefits of XRT though in terms of speed and accuracy.

No doubt XRT (or MCT) is faster and more accurate than SEM determination, and gives data that can be treated statistically. Here we have not explained properly the benefits of our methodology, which is a fail in this section, especially when we have worked in the comparison between SEM and MCT representativeness. We modify the text to explain this point. Please see lines 51-58

L70 new framboids from old.  This should be established for the samples you are looking at and they should be documented.  Granted it may only be morphology but this is useful information.

“In a recent paper [14] we have shown that low-grade metamorphism prompts the development of new populations of framboids in pelitic rocks as metamorphic conditions change”. This means that during metamorphism, new framboids are formed but not from the original framboids formed under sedimentary conditions. In fact, it is impossible to differentiate sedimentary from metamorphic framboids as it is explained in the referenced paper number 14.

L75 The sources are indeed from two distinct rock types.  This should be better documented in results section.

We rewrite the manuscript, separating samples by terrane. Results and discussion are now referred to terranes instead of the “Otago formation”

L81 There are two distinct lithologies here.  Further, greywacke is a rather generic term.

In accordance with the previous comment, we make reference to these two distinct lithologies in the paper. We prefer to maintain the term “greywacke” since Pitcairn et al. 2010 uses this term. We have used the same samples from this paper, so it could be confusing to change terminology.

Pitcairn, I.K.; Olivo, G.R.; Teagle, D.A.H.; Craw, D. Sulfide Evolution during Prograde Metamorphism of the Otago and Alpine Schists, New Zealand. The Canadian Mineralogist 2010, 48, 1267-1295, doi:10.3749/canmin.48.5.1267.

L117 There are a lot of software packages mentioned here.  Would suggest putting their names in italics or something to allow their recognition.

Done

L118 setup?  Facility?

We change setup by facility

L120 this description …

We delete that part of the sentence

L121 I’m not an expert in XRT so why is the average voltage distinct from peak?

Each scan has different parts (calibration, image acquisition) and tube voltage oscillates for each stage. Also each sample has different parameters (density, number of objects). However, this part of the sentence might be confusing for a person not familiar with XRT (MCT), and it does not add any information, so we prefer just delete it.

L131 I think the morphologies are key to making this a good paper.

With the MCT technique only spherical morphologies can be detected and measured. More detailed morphologies need other advanced technique to be studied. The objective of the study is to obtain firstly spherical morphologies from the MCT images and then populations of framboids from the mixture of log-normal populations. This objective is reflected in results and discussion and used to establish the transition from diagenetic to metamorphic conditions when a new framboid population is detected.

L138 Samples show diversity.  This must be explained and the samples properly documented otherwise there is no point to this paper.

The diversity of the samples is explained in the new version of the text. Also, these samples are exhaustively analyzed in Pitcarin et al. 2010, this is the reason we do not go deep into this.

L151 recrystallisation: ilmenite - titanite is not a simple recrystallisation.  it is a metamorphic transformation involving movement of Si, Fe and Ca (for a start).

Indeed the process is not correctly explained, we change from recrystallization to being replaced (the other reviewer also pointed this out).

L170  Again, these “recrystallisations” are the result of chemical reactions

We change metamorphic recrystallization to metamorphic replacement

L179 Figure 2 is rather uninformative without descriptions of the parameters

We have redrawn this figure, and also rewritten captions

L181 Figure 3 does indeed show framboids in (a) and (b).  Interesting that (b) appears to be structured suggesting Oswald ripening.  But what happens at higher grade?  Is there a morphological shift?  What is the point of (c)? Simply the recognition of pyrite in the algorithm? (d) interesting that the algorithm seeks to put everything in to euhedral crystals.  The framboid certainly doesn’t look like this.  And we see no comparator for the euhedral pyrite.

The point of this work is the development of new MPy populations, discriminated by their size. When the metamorphism increases, populations tends to be composed by non-spherical objects, what we call unrecognizable objects. In (c) we show how the MPy are seen in a MCT scan, to give the reader an overview of the technique. In (d), reconstructed images might look like euhedral crystals, but this is an effect of the resolution of the scan. We are working at a very high resolution, 3 µm voxel. Framboids are recognized by their geometrical values, not because they look like a framboid. Regarding the euhedral pyrite, these reconstructions are merely illustrative, our results come from the datasheets.

Figure 3 (b) represents a recrystallizated framboid of pyrite. This kind of transformation is probably achieved under sedimentary conditions and not during metamorphism. It’s included here to show different kind of morphologies of the studied samples.

Previous works have shown that this type of recrystallizations (sunflowers) can be created under many different environments, not only sedimentary, e.g. Merinero & Cardenes 2018, Merinero et al. 2015.

Merinero, R.; Cárdenes, V., Theoretical growth of framboidal and sunflower pyrite using the R-package frambgrowth. Mineralogy and Petrology 2018, 112, 577-589.

Merinero, R.; Cárdenes, V.; Lunar, R.; Boone, M. N.; Cnudde, V., Representative size distributions of framboidal, euhedral, and sunflower pyrite from high-resolution X-ray tomography and scanning electron microscopy analyses. American Mineralogist 2017, 102, 620-631.

At higher grade of metamorphism, framboids of pyrite are not formed and probably dissolved it.

This is true, as it can be seen in Figure 4, the higher grade samples are not framboids.

Figure 3 (c) shows how the MCT technique differentiate pyrite objects from the matrix of the sample. After the morphological filter, only spherical objects are considered (see details of the technique in the manuscript)

Our methodology has considered other morphologies than spherical, as it can be seen in Figure 4

Figure 3 (d) shows the reconstruction of typical spherical objects: framboids and euhedral crystals. This theoretical 3D reconstruction is put here to show the final results of the filtering process where spherical objects are obtained from the whole pyrite objects differentiate with MCT technique.

As pointed before, this is just to show the reader how objects are reconstructed by the software, but always having in mind the resolution.

L189 metamorphism … ran its course.  Not sure what this means.

Not appropriate English, we change to “metamorphism progrades”

L192 … increase in metamorphic degree … needs to be more specific.

In accordance with previous comment, we change to “The progradation of metamorphism…” see lines 202-203

L194 “regular synkinematic textures”  euhedral? rotated?

We include “rotated” in the description

L196 unrecognisable shapes (probably euhedral)… So euhedral shapes are not recognisable?  Look at the samples.

Some subpopulations are composed by object with no clear shapes. This lack of recognition is due to the limitation of MCT object analysis. Statistical analysis is able to separate the different subpopulations but not depending on the shapes, but in their adjustment to the log-normal distribution. Also, we proceed to change “unrecognizable shapes (probably euhedral)” by “other spherical shapes (euhedral)”

L205 there is not a recognisable morphology …  No.  Morphology can be described.

We change “recognizable” to “describable”

L208 We need more documentation of these unrecognisable features.

The determination of the morphological features was performed using the previously mentioned ImageJ plugins. This information is given as size distributions. Non-recognizable objects are those which are not of spherical or euhedral shape.

L211 Figure 4. I don’t understand the symbol distinctions here.  Looks like everything is either a framboid or a framboid cluster.  No formation of epigenetic pyrite?  There is no interpretation of the data on this plot.  Is the correlation between size and SD meaningful in any way? Is there a difference between Torlesse and Caples?

This graphic is the classical way to represent size population analysis. It is used here to give the reader a chance to correlate the information provided in this paper with previous works. Because of that, we think it is important to include it. Into this graphic, we have included a sub-graphic to differentiate between Caples and Torlesse terrains.

L217 if the framboids disappear, where does the Fe S go?  Note in Fig 3, sulphides are apparent in amphibolite grade.

In this work, we analyzed the morphologies of sulphides and the lack of spherical sulphides (framboids) during the prograde of metamorphism. The object of this work is not another geochemical study of these rocks, but the occurrence of MPy populations and its relation with the metamorphism. Figure 3 represents sample A6 unmetamorphosed greywacke with framboidal pyrite.

L222 There is little discussion concerning the relationship between the size/SD and the mineralogy/morphology

Figure 4 is used here as reference of the variation in mean and sd of the different populations of sulphides detected during the analysis of samples. Probably no relationship between mean and sd can be established in the studied samples, but the formation of the new populations of framboids can be displayed. Framboids are always formed under oxic conditions during metamorphism, and probably other spherical shapes as euhedral crystals of pyrite were formed under anoxic conditions previously to the metamorphism, but not evidence of this can be established with the study of morphologies of sulphides with MCT. We include this on the discussion.

Round 2

Reviewer 2 Report

I'm satisfied that the authors have made a good attempt to address all my concerns and observations and that the paper should be accepted.  

For this round of review, I limit my comments to reading the manuscript for consistency, English etc.

L29 magnetite is not a sulfide... maybe rephrase a touch.

L46,48 repetition of isotopic and chemical analysis at start and end of sentence

L165 did not showed (replace with "did not show" ?)

Author Response

L29 magnetite is not a sulfide... maybe rephrase a touch.

Response: correct, we delete magnetite

L46,48 repetition of isotopic and chemical analysis at start and end of sentence

Response: we delete the last part of the sentence

L165 did not showed (replace with "did not show" ?)

Response: done
